# New Distributional Records and Characterization of the Climatic Niche of *Lepturges* (*Lepturges*) *limpidus* Bates, 1872 (Coleoptera, Cerambycidae): Sink or Source Population?

**DOI:** 10.3390/insects13111069

**Published:** 2022-11-18

**Authors:** Néstor G. Valle, Marianna V. P. Simões

**Affiliations:** 1Facultad de Ciencias Exactas y Naturales y Agrimensura, Universidad Nacional del Nordeste, Avda. Libertad 5470, Corrientes W3400 BCH, Argentina; 2Senckenberg Deutsches Entomologisches Institut, Eberswalder Straße 90, 15374 Müncheberg, Germany

**Keywords:** ecological niche modeling, niche overlap, Neotropics, ellipsoid envelope, new distributional record, host plant

## Abstract

**Simple Summary:**

Longhorned beetles (Coleoptera: Cerambycidae) are one of the most diverse, ecologically and economically important groups of beetles in the world. In the Neotropical Region, more than 8000 species are known and new records are added each year, linked to increased international trade, reduced transport times, and an increase in the number of potential vectors. In this study, we report *Lepturges* (*Lepturges*) *limpidus* as an exotic species in Argentina and discuss the potential drivers of its expansion through modeling to help understand and predict the possible geographic distributions of the species as a function of environmental factors. In addition, in view of the results obtained, we discuss the relevance of biomonitoring programs and field studies to detect the arrival of potential invasive alien species in the region.

**Abstract:**

A growing number of cases of the spread and establishment of non-native species outside their previously known ranges has been reported in recent years. Here we report new distributional records of *Lepturges* (*Lepturges*) *limpidus* Bates, 1872 (Cerambycidae) from Argentina and investigate whether these records could represent established populations. We constructed ellipsoid envelope models to characterize climatic niches of *L. limpidus*, identified areas of climatic suitability, investigated the status of new records as climatic outliers, and evaluated its dependency on its known hostplant as a limiting factor for the beetle distribution. Results indicate widespread climatic suitability in the Neotropical Region, and new records are not outliers with regard to the climatic profile of *L. limpidus*. Association with its known hostplant is non-dependent, indicating that the species might utilize different hosts plants. New records likely represent established populations, but targeted surveys should be carried out to detect new arrivals and enable the installation of mitigation and control measures.

## 1. Introduction

Longhorn beetles (Coleoptera: Cerambycidae) are one of the largest, most diverse, ecologically and economically important groups of beetles [1]. The family comprises more than 38,000 described species worldwide, which are all phytophagous, xylophagous or saproxylophagous [2]. Their larvae consume a great variety of plant parts and tissues, in diverse conditions, from alive and healthy to dead and decaying [3]. Adaptation to this large spectrum of host plant conditions has resulted in great variation in the behavior and ecology of these beetles, whereby many species are considered important pests of agricultural crops, ornamental trees, forestry, and timber products [3,4,5].

In the Neotropical Region, more than 8000 species are known [6], with a growing number of new distributional records each year (e.g., [7,8,9,10,11]). Such expansion has been linked to increased international trade, reduction of the duration of transport and the increase in diversity of potential vectors [12,13,14,15,16], altogether aiding the spread and establishment of non-native species outside their native ranges [17]. A small number have been intercepted outside their native ranges, and some of them have established new populations that cause serious ecological and economic damage (e.g., Asian Long-horned beetle, *Anoplophora glabripennis*; [3,18,19]). Thus, it is strategically relevant to report and understand drivers of the range expansion of such species.

Climate-matching approaches, a key concept of which is the linkage of species distributions and environmental conditions, can assist in the exploration and forecasting of species range expansion in response to changing environments [20,21]. Insights gained from such analyses can help elucidate current patterns and provide a first approximation of changes in distributional patterns due to anthropogenic climate change [22]. However, while environmental conditions have been the main predictors employed to understand coarse-scale patterns, the range of species is also determined by other factors operating at finer-scales, such as biotic interactions and movement or dispersal factors [23]. Biotic interactions for instance, may be restrictive for phytophagous species, while the presence of their host plant is essential for successful development of larvae and establishment of a population at a particular site [24,25,26,27,28,29]. The interaction of these, along with dispersal capability [23], help to identify habitat patches that support local population growth and serve as net exporters of individuals (i.e., source populations), or habitat patches where mortality exceeds natality and thus cannot sustain local populations (i.e., sink populations; [30]). Thus, the characterization of populations as source or sink could aid the identification of newly established populations and thereafter help to refine strategic policy in pest eradication campaigns.

The longhorn-beetle *Lepturges* (*Lepturges*) *limpidus* Bates, 1872 (Cerambycidae) is an exclusively Neotropical species, distributed from central Mexico to southern Paraguay [6]. Recently, fieldwork survey performed by the first author (NV) in November 2016 and March 2021 revealed new distributional records for this longhorn beetle species in the northeast of Argentina. Located within the Humid Chaco ecoregion, the region shows a mosaic of ecosystems including woodland and savanna, with various species of trees, shrubs, and coarse grasses [31]. This vegetation and environmental profile is similar to those within the previously known distribution of *L. limpidus*. However, despite extensive sampling in previous studies in that area (e.g., [32,33,34,35]) and similar phytophysiognomy and climate, these are the first records of the longhorn beetle species from Argentina. Additional sparse information is available regarding the biology and ecology of the species, such as its only known hostplant association, with *Catostemma fragrans* Benth. (Malvaceae) [6].

As a first step towards predicting and preventing invasion, we seek to understand whether the new records in the northeast of Argentina could represent an established population of *L. limpidus*. With that aim, we (i) projected ellipsoid models onto geographic spaces to identify areas of climatic suitability, (ii) tested the closeness of new records to the climate optimum, and (iii) measured the level of niche overlap based on ellipsoid envelopes between *L. limpidus* and its hostplant species, to test the dependency of their interaction as a limiting factor for the distribution of the beetle. In addition, we report the new distributional records of *L. limpidus* in Argentina, discuss the potential drivers of range expansion and whether new records could represent established (i.e., part of a source population) or spurious records (i.e., part of a sink population), in the light of the ellipsoid envelope results.

## 2. Materials and Methods

*Study area*. The specimens were collected in the city of Colonia Benítez, Chaco Province, Argentina. The area is included in the Chacoan biogeographic province [36], which has a subtropical climate, with an average annual temperature of 21 °C (minimum −3 °C and maximum 44 °C). The annual mean precipitation is 1300 mm, concentrated mainly in summer [37].

*Distribution records*. In total, 98 occurrence records for *Lepturges* (*Lepturges*) *limpidus* were assembled: 15 records extracted from the literature ([6,38,39,40,41,42,43,44,45,46,47,48]); 41 records obtained from the Global Biodiversity Information Facility (GBIF; from GBIF Occurrence Download https://doi.org/10.15468/dl.hmjbr5 (accessed on 1 April 2022)), and one record from Specieslink (http://www.splink.cria.org.br/ (accessed on 1 April 2022)) (Table 1). The sampling was complemented with 41 records obtained from museum collections: Museu de Zoologia de Sao Paulo (MZSP); Universidad Federal do Paraná (DZUP, Coleção Entomológica Pe. Jesus Santiago Moure); Museo Argentino de Ciencia Naturales “Bernardino Rivadavia” (MACN); Smithsonian National Museum of Natural History (USNM) and Museo Javeriano de Historia Natural (MPUJ). The known native distribution of longhorn beetles was defined based on the catalog of [6] and the online database of Cerambycidae by [2]. Locations lacking geographic coordinates were georeferenced using Google Earth (https://www.google.com/earth/ (accessed on 1 April 2022)) and Global Gazetteer (http://www.fallingrain.com/world/ (accessed on 1 April 2022)).

For the only known hostplant *Catostemma fragrans*, we assembled 177 occurrence records: 103 occurrences obtained from GBIF (GBIF; Occurrence Download https://doi.org/10.15468/dl.6g2usj (accessed on 1 April 2022)) and 74 occurrences obtained from speciesLink (https://specieslink.net/search/download/20220405081840-0011473 (accessed on 1 April 2022)). Its native distribution was cross-checked against Plants of the World Online (POWO) (http://powo.science.kew.org (accessed on 1 April 2022)). The following museum and herbarium collections were consulted: Field Museum of Natural History; Naturalis Biodiversity Center; New York Botanical Garden; United States National Herbarium; but no records were obtained. Distribution records were visually inspected for adequacy, and dubious records were corrected (e.g., reversed latitude and longitude fields), or removed following the protocol in [49]. To avoid problems derived from spatial autocorrelation and to reduce sampling bias, a thinning distance was chosen to take account of the spatial resolution of variables (~9.2 km at the equator), and the effect on geographic clustering and effective number of remaining points after exploring shorter and longer distance alternatives (i.e., 5 km, 10 km, 50 km). Accordingly, the number of occurrence records was rarefied by spatial thinning using a 10 km distance, resulting in a final total of 42 included records for *L. limpidus* (Table 1; Figure 1A) and 28 records for *C. fragrans*. These records were used to calibrate and create the final models. To clean and analyze the data, all steps were performed using the statistical software RStudio, version 4.2.0, (RStudio team, Boston, MA, USA) [50] and packages ‘spThin’ [51], ‘raster’ [52], and ‘rgdal’ [53].

New records were obtained during field work performed by the first author (NV), in November 2016 and March 2021. These specimens were deposited at Universidad Nacional del Nordeste, Corrientes, Argentina (CARTROUNNE). Species identification was based on comparison of collected specimens with images in the photographic catalog of the Cerambycidae of the New World [54] and following descriptions provided by [42]. Finally, the identifications were confirmed by the specialist Dra. Marcela Monné (Museo Nacional/UFRJ, Rio de Janeiro, Brazil). The new records from Argentina were not included in the modeling analysis.

*Environmental data*. Environmental data for this study were obtained at 2.5 arc-minute (~4.6 km at the equator) spatial resolution from WorldClim (version 1.4, http://www.worldclim.org (accessed on 1 April 2022) [55]). WorldClim is based on interpolations of weather station data (i.e., monthly precipitation and minimum and maximum temperatures) over the period 1950–2000. From the 19 variables available, we excluded four (mean temperature of wettest quarter, mean temperature of driest quarter, precipitation of warmest quarter, precipitation of coldest quarter) a priori due to known spatial artifacts between adjacent grid cells [56,57]. To avoid overfitting, overly dimensional environmental space, and collinearity among variables, we cropped the variables to the extent of the Neotropical Region and performed principal component analysis (PCA) using the function kuenm_rpca in the package ‘kuenm’ [58] in R, version 4.0.3 (R Core Team, Vienna, Austria) [59]. To build the ellipsoid niche model of *L. limpidus* we retained the first three components for model calibration. These explained cumulatively > 88% of the total variance in the dataset (see Appendix A). Following [60], to calculate the ellipsoid niche overlap between *L. limpidus* and *C. fragrans*, we tested three distinct environmental sets seeking to avoid bias regarding the combination of variables used to characterize the species niche centrality: ‘set 1’ included all 15 variables; ‘set 2’ included only temperature variables (i.e., annual mean temperature; mean diurnal range; isothermality; temperature seasonality; max. temperature of warmest month; min. temperature of coldest month; temperature annual range; mean temperature of warmest quarter and mean temperature of coldest quarter); and ‘set 3’ included only precipitation variables (i.e., annual precipitation; precipitation of wettest month; precipitation of driest month; precipitation seasonality; precipitation of wettest quarter; precipitation of driest quarter). As a result, we retained the first three components of each set, which explained cumulatively > 90% of the total variance in the dataset for model calibration (see Appendix A).

*Ellipsoid niche model*. To characterize the environmental niches of *L. limpidus* and C. fragans we used ellipsoids as climatic niche models. Ellipsoid niches have been hypothesized to provide a close approximation to a fundamental niche [61,62,63] and assume that the ecological niche of a species has only one optimum (i.e., niche centroid; see [62,64,65]). Therefore, this is an optimal method for interpreting the multidimensional parameter space of habitat variation and roughly categorizing records as belonging to source or sink populations [60].

Models were built using the ‘ellipsenm’ package [66], calibrated using the 95% pairwise confidence region for the ellipsoid, and evaluated using the function ‘ellipsoid_calibration’ [66]. Two different methods were employed to construct ellipsoid models: (1) ‘covmat’, which creates ellipsoids based on the centroid and a matrix of co-variances of the variables and (2) ‘mve1’, which generates an ellipsoid that reduces the volume contained in it without losing the data contained (i.e., minimum volume ellipsoid; [67]). Best-model selection was based on statistical significance (partial ROC; [68]); the proportion of testing data known to be in suitable areas and prediction of unsuitable areas was based on omission rates (E = 5%; [69]) and prevalence (i.e., proportion of space identified as suitable for the species; [66]). To calculate the partial ROC metric, we used 500 bootstrap iterations with 50% of testing data to be used in each bootstrapped process with 5% of testing data error due to uncertainty. Prevalence was calculated in geographical and environmental space. In geographic space, proportions were calculated using all pixels from the raster, while in environmental space only pixels that have distinct combinations of all variable values were considered [63,66]. The calibration area (i.e., region accessible to the species; [70]) included a buffer of 50 km from the occurrence records included in our models. The buffer size was defined based on the known dispersal ability of species of longhorn beetle subfamily Lamiinae [71] and on previous studies on the distribution of the hostplant (i.e., [72,73]). Final parameters were selected based on the best evaluated models and used to create the final models using 10 replicates with bootstrapped subsamples of 75% of the data using the function ‘ellipsoid_model’ available in the ‘ellipsenm’ R package [66]. The replicates were produced by excluding one occurrence record at a time. We represented ecological niche and suitability levels of *L. limpidus* in geographic space, which were binarized using a threshold for suitability aiming to exclude the 5% of the data with the most extreme values. Visualization of results was performed in QGIS (version 3.10, QGIS Development Team, http://www.qgis.org (accessed on 22 July 2022) [74]).

*Spatial error analysis*. Suitability values in ellipsoid envelope models represent the Mahalanobis distance from the optimum (i.e., ellipsoid centroid), while maximum values will be close to the centroid and minimum values will be close to the periphery of the ellipsoid [66]. Thus, to verify whether the new records in Northern Argentina represent outliers in the environmental space, we extracted values for all known occurrence points of *L. limpidus* from its predicted environmental suitability, using the mean of replicates. Then we visually inspected the position of the new records within the distribution of Mahalanobis distance values of records of the known distribution of *L. limpidus*.

*Ellipsoid niche overlap*. To test whether the hostplant might be constraining *L. limpidus* distribution, ellipsoid niche overlap was calculated considering the union of the environmental conditions relevant for both species, *L. limpidus* and *C. fragans*. The process was repeated 1000 times, and the observed overlap values were compared to the values found for pairs of random ellipsoids. The null hypothesis is that the niche of *L. limpidus* is contained within the *C. fragans* ellipsoid, and if the observed values are as extreme or more extreme than the lower limit of the values found for the random ellipsoids, the null hypothesis is rejected. That is, if the observed overlap is lower than the 95% of the random-derived values of overlap, the niches are considered non-overlapped. If the observed values cannot be distinguished from the random, the null hypothesis cannot be rejected. A *p*-value and the predefined confidence limit is added to the overlap matrix when the test is performed.

## 3. Results

*New distributional records*. *Lepturges limpidus* (Figure 1B) is widely distributed in the Neotropical Region and occurs in Mexico (Jalisco, Veracruz), Guatemala, Honduras, El Salvador, Nicaragua, Costa Rica, Panama, Colombia, Brazil (Pará, Mato Grosso, Espírito Santo, São Paulo, Paraná, Santa Catarina), Peru, Bolivia (Beni, Santa Cruz, Tarija), and Paraguay [6]. New country record for Argentina (Department Primero de Mayo) (Figure 1C,D).

*Examined material*. Body length: male = 7.25–7.74 mm (mean = 7.50 mm); width = 2.13–2.36 mm (mean = 2.25 mm); female = 6.47–7.74 mm (mean = 7.11 mm); width = 1.84–2.47 mm (mean = 2.16 mm). ARGENTINA: Chaco: Colonia Benítez, Los Chaguares, Department Primero de Mayo, 27°19′59″ S; 58°57′57″ W, 55 m above sea level, 3 males and 1 female, 30-XI-2016, N. G. Valle leg. (CARTROUNNE 9357; 9495; 9496; 9497); Private field, 27°20′17″ S 58°58′01″ W, 1 female, 12-III-2021, N. G. Valle leg. (CARTROUNNE 9498).

*Ellipsoid niche model of L. limpidus*. The geographic projections of the ecological niche of *L. limpidus* showed widespread suitable areas across the Neotropics, including northern and central Argentina. Specifically, climatic suitability was widespread within the Chaco biogeographic province and concentrated mainly on the Eastern Chaco district, where new records for *L. limpidus* are located (Figure 1A). The best fitting method to construct the climatic ellipsoids was ‘covmat’; mean AUC, *p*-value of partial ROC and omission rates were significantly better than random expectations (*p*-value < 0.05; Table 2). Prevalence of mean ellipsoidal models in geographical (G-space) and environmental (E-space) space were relatively high (0.712; Table 2). The complete specifications of ellipsoid characteristics (e.g., centroid, covariance matrix, semi-axes length, etc.) can be found in Appendix A.

*Spatial error analysis*. Mahalanobis distance values (MD) recovered for new records in northeast Argentina were low: 0.043 for Private field and 0.046 for Los Chaguares. Visual inspection revealed that new records are close to the periphery of the ellipsoid envelope of *L. limpidus* (Figure 2).

*Ellipsoid niche overlap of L. limpidus and C. fragans*. We rejected the hypothesis that *L. limpidus* is contained within the niche of *C. fragans* (see *p*-values in Table 3, Figure 3). All combinations of bioclimatic variables revealed that the longhorn beetle has ca. three times the hypervolume of the hostplant *C. fragans* ellipsoid. Niche volumes were similar between these species when only precipitation or temperature variables were considered, but when all variables were included, niche volume of *L. limpidus* increased substantially, while the volume of *C. fragans* reduced in size (Table 3).

## 4. Discussion

Species distributions are conceived as the intersection of three limiting factors: movement capacities, abiotic conditions, and biotic interactions [23]. The relative significance of these components is influenced by spatial scale and resolution [24,25,75]. Recent fieldwork revealed new distributional records for the longhorn beetle *Lepturges (L.) limpidus* Bates, 1872 (Cerambycidae) in northern Argentina. The beetle species shows a wide distribution throughout the Neotropical Region, but this is the first record for the Chaco biogeographic province. Our results indicate widespread climatic suitability within the Neotropical Region, but new records in northeastern Argentina are near the periphery of the *L. limpidus* ellipsoid envelope. Further, based on the ellipsoid envelopes, no dependent association to its known hostplant *C. fragans* was recovered.

A central postulate in biogeography is that climate exerts a dominant influence on the distribution of species (e.g., [76,77,78,79]). For *L. limpidus,* climatic niche projections revealed widespread suitable areas within the Neotropics, with high geographic (i.e., proportion of total suitable area) and environmental space prevalence (i.e., proportion of the multidimensional space identified to be suitable considering only distinct combinations of environmental conditions). This was particularly observed within the eastern part of the Chaco biogeographic province (Eastern Chaco district), characterized by a humid subtropical climate without a marked dry season and by a hot summer season [37], favorable for the development of different cerambycid species and the growth of their potential hosts [80,81,82]. However, Mahalanobis distance values associated with occurrences in northeast Argentina were relatively low (MD = 0.043 and 0.046; Figure 2), suggesting that these records are probably attributable to climatically peripheral populations.

Moreover, association with the climatic space of its hostplant *C. fragrans* was non-dependent. As a native species of the Guianas [83], *C. fragrans* is intensively harvested and traded as a commercial timber tree [84,85], which could have potentially facilitated the translocation (i.e., accidental or intentional human-mediated movement[s] of living organisms from one area, with release in another; IUCN 2012) of *L. limpidus* to northeastern Argentina. The endophytic lifestyle of cerambycid species allows their easy and widespread movement through the international trade in timber and timber derivatives [86,87,88]. Unintentional transport of species within or on a specific commodity is frequently reported. In Europe, for instance, this accounts for ca. 90% of alien arthropod species invasions [89]. A particular case is the polyphagous invasive beetle *Anoplophora glabripennis* (Motschulsky, 1854) native to Southeast Asia, which threatens forests in Europe and North America and spreads in wooden packaging materials (boxes/pallets) and their derivatives, managing to adapt to new territories due to its wide range of hostplants [90]. Thus far, evidence suggests that the new records of *L. limpidus* are a result of translocation, potentially representing sink populations, while biotic and abiotic interactions in the new distributional area seem sub-optimal for the establishment of populations.

Nevertheless, it is hard to ascertain the status of a population as sink or source. Local adaptation of a translocated species to a new geographic area can occur due to its ability to exploit empty niches in the ranges to which it is introduced [91], or the frequency and magnitude of local processes (e.g., disturbance), and the absence of its natural enemies, including competitors, predators or pathogens [92,93,94,95,96]. Likewise, translocations may enhance evolutionary changes partly owing to founder effects and genetic bottlenecks and partly because of the triggering of evolution by new environmental factors [97]. *Zygogramma suturalis* (Fabricius, 1775) (Coleoptera: Chrysomelidae) for instance, showed rapid evolutionary changes in flight capacity (development of flight ability and morphological changes) within only five generations [98].

Then again, *L. limpidus* could alternatively be utilizing other hostplant species as resources; however, our knowledge regarding additional biotic interactions is inadequate. Despite great economic relevance and prominence in ecology and evolutionary literature, the biotic interactions of xylophagous insects and the essence of their interaction (e.g., hostplants as food source, oviposition sites, or shelter for hibernation) remain poorly known [29,99]. This deficit hinders the development of targeted monitoring programs for species at high risk of becoming invasive pests and the protection of areas that could represent areas of potential distribution for pest species [29]. Among xylophagous insects, most cerambycid species present different specialization strategies, occupying various host plant species [100]. As expected in such a large insect family, some cerambycids are strictly monophagous, while others develop in multiple plant species of the same family (oligophagous) or multiple families (polyphagous) [71,101]. Also, some cerambycids infest living, healthy plants, while others develop in dying plants; likewise, some species prefer moist wood, while others prefer dry wood [3]. In the Neotropics, [43] determined the pattern of host specificity for cerambycids from the tropical rainforest of French Guiana, where almost 24% of the species had a single recorded host, while 53% were generalists, mostly Acanthoderini or Acanthocinini. Alternatively, a new record based on only a few specimens could represent a recent colonization (e.g., the three males and two females of *L. limpidus* from northeast Argentina). However, as in many cases involving arthropod species, the introduction of the species cannot be ascertained [102], and the small number of specimens might reflect a lack of target sampling or incongruency between species biology and field surveys. Whatever the case, further monitoring surveys should be implemented to detect, document and investigate potential establishment of *L. limpidus* in the area.

Finally, it is important to highlight that while our models help to understand and predict potential geographic distributions of species based on environmental factors, they do so without considering important factors of microenvironments and vegetation cover. We cannot overlook that cerambycid species are often closely linked to forest environments [71], so including vegetation cover layers (e.g., normalized difference vegetation index, NDVI) in future studies could contribute to a broader interpretation of the species’ distributional model [29,103,104].

## 5. Conclusions

In summary, we report new records of *L. limpidus* in Argentina, and our results show that its new distributional record possibly represents a sink population, potentially resulting from translocation associated with the increased commercialization of its hostplant as timber. Although Chaco is the second-most forested area in South America [105], it has undergone extensive land-use changes that have led to significant deforestation [106,107]. Hence, target surveys are essential for the detection of new arrivals of exotic species, their patterns of invasion, and exotic species impacts to allow installation of mitigation and control measures.

## Figures and Tables

**Figure 1 insects-13-01069-f001:**
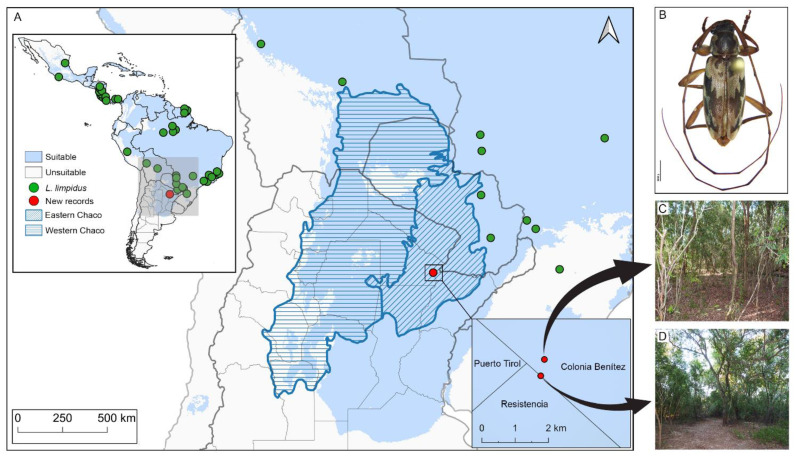
Climate suitability map and distribution of *Lepturges limpidus*. (**A**) Map shows climatic suitability recovered for *Lepturges limpidus* within biogeographic provinces of South America. The oblique blue lines indicate the Humid Chaco ecoregion. Green points are known distribution and red points new records reported in the present paper; (**B**) Dorsal view of *Lepturges limpidus*, (**C**) Private field, (**D**) Los Chaguares.

**Figure 2 insects-13-01069-f002:**
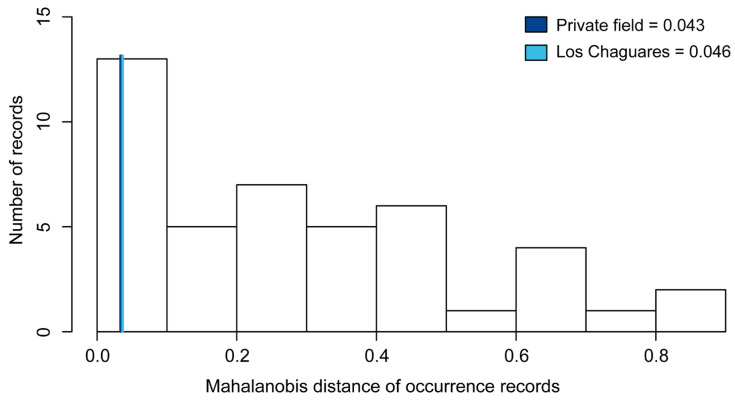
Histogram shows Mahalanobis distance of occurrence records to the ellipsoid centroid of *Lepturges limpidus*. Values associated with the new distributional records reported in the present study are indicated in blues lines, Line dark blue represents “Private field” and Line light blue represents “Los Chaguares”.

**Figure 3 insects-13-01069-f003:**
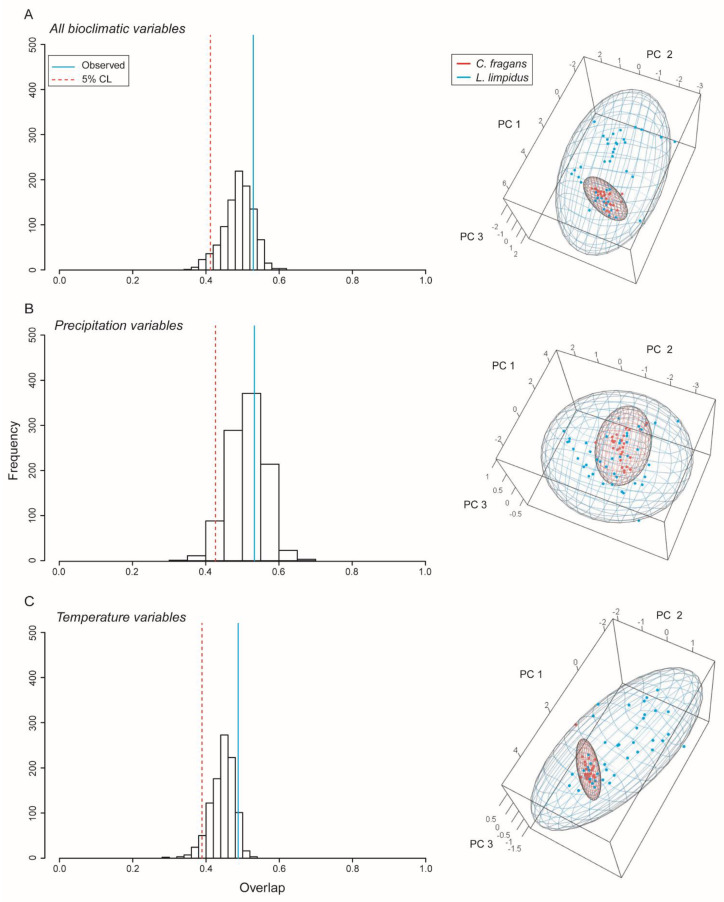
Niche overlap between the longhorn beetle *Lepturges limpidus* and its known hostplant and *Catostemma fragrans.* Blue and red points represent occurrences for *L. limpidus* and *C. fragans*, respectively. Histograms show overlap and significance (**left**) and ellipsoid models indicating overlap in environmental space (**right**). (**A**) Results using all bioclimatic variables, (**B**) Precipitation variables, and (**C**) temperature variables.

**Table 1 insects-13-01069-t001:** Known records of *Lepturges limpidus* in the Neotropics sorted by locality, country and geographic coordinates.

Locality Name	Latitude	Longitude	Reference
Itatiaia, Rio de Janeiro (Brazil)	22°29′46″ S	44°33′47″ W	speciesLink (2022)
Valentim Gentil, Sao Paulo (Brazil)	20°22′25″ S	50°05′17″ W	GBIF.org (2022)
Céu Azul, Parana (Brazil)	25°04′09″ S	53°39′35″ W	GBIF.org (2022)
Laranja da Terra, Río Guandú, Espirito Santo (Brazil)	19°54′39″ S	41°05′04″ W	[41]
Parintins, Amazonas (Brazil)	02°38′06″ S	56°43′55″ W	MZSP
Rio Purus, Amazonas (Brazil)	03°41′20″ S	61°26′46″ W	MZSP
Maués, Amazonas (Brazil)	03°23′00″ S	57°43′06″ W	MZSP
Salobra, Mato Grosso (Brazil)	20°11′59″ S	56°31′39″ W	MZSP
Fazenda Beija Flor, Mato Grosso (Brazil)	21°02′06″ S	56°27′23″ W	MZSP
Nova Teutônia, Santa Catarina (Brazil)	27°09′48″ S	52°25′22″ W	MZSP
Linhares, Espírito Santo (Brazil)	19°23′57″ S	40°03′56″ W	MZSP
Córrego do Itá, Espírito Santo (Brazil)	18°38′21″ S	40°51′42″ W	MZSP
Rio Nhamundá, Pará (Brazil)	01°10′09″ S	57°57′51″ W	MZSP
Pouso Alegre, Minas Gerais (Brazil)	22°13′41″ S	45°56′01″ W	MZSP
Belo Horizonte, Minas Gerais (Brazil)	19°55′41″ S	43°56′31″ W	MZSP
Viçosa, Minas Gerais (Brazil)	20°45′14″ S	42°52′55″ W	MZSP
Rondon, Brasilien (Brazil)	24°38′00″ S	54°07′00″ W	Smithsonian Institute
Hotel F & F, Buena Vista, Santa Cruz (Bolivia)	17°27′31″ S	63°40′09″ W	Smithsonian Institute
Guanay (Bolivia)	15°29′54″ S	67°53′03″ W	MZSP
Area de Conservación Guanacaste, La Cruz, Finca Jenny, Guanacaste (Costa Rica)	10°51′57″ N	85°34′26″W	GBIF.org (2022)
Bagaces, Parque Nacional Palo Verde, Sector Palo Verde, Guanacaste (Costa Rica)	10°20′56″ N	85°21′08″ W	GBIF.org (2022)
A.C.P.C, Garabito, Tarcoles, Estación Quebrada Bonita, Puntarenas (Costa Rica)	09°46′02″ N	84°36′29″ W	GBIF.org (2022)
Osa, Ciudad Puerto Cortes, Cuesta del Burro Puntarenas (Costa Rica)	09°01′25″ N	83°30′31″ W	GBIF.org (2022)
Ebene Limón, Reventazon, Hamburg Farm (Costa Rica)	10°04′45″ N	83°34′39″ W	MZSP
Sándalo, Golfo Dulce (Costa Rica)	08°34′08″ N	83°22′14″ W	Smithsonian Institute
Zone Sinnamary, Crique Plomb, Sinnamary (French Guyana)	05°00′00″ N	52°57′14″ W	[43]
Zone Bélizon, Route Forestière, Roura (French Guyana)	04°16′33″ N	52°38′34″ W	[46]
Zone L’île de Cayenne, Rémire (Degrad des Cannes), Cayenne, (French Guyana)	04°53′02″ N	52°19′12″ W	[46]
Zone Iracoubo, RN 1 (PK 172), Iracoubo (French Guyana)	05°29′20″ N	53°19′58″ W	[46]
Zone Centrale, Saül, Saül (French Guyana)	03°51′57″N	53°23′13″ W	[46]
El Paraiso, Caripe, (Honduras)	13°58′55″N	85°49′26″ W	[44]
NE Ixtapa, Hwy 200, Guerro (Mexico)	17°39′28″N	101°34′32″ W	Smithsonian Institute
Gomez Farias, Bocatoma, Tamaulipas (Mexico)	22°59′15″ N	99°08′55″ W	Smithsonian Institute
Barro Colorado I., C. Zone (Panamá)	09°09′58″ N	79°50′21″ W	Smithsonian Institute
W. Ipiti, Bayano (Panamá)	09°09′00″ N	78°50′00″ W	Smithsonian Institute
Los Guatuzos, Rio Papaturro, Río San Juan (Nicaragua)	11°02′27″ N	85°05′13″ W	GBIF.org (2022)
Las Flores, Masaya (Nicaragua)	12°00′16″ N	86°01′11″ W	GBIF.org (2022)
Matagalpa, La Sombra (Nicaragua)	13°11′06″ N	85°45′00″ W	[47]
Chontales, (Nicaragua)	12°16′00″ N	84°59′00″ W	[38]
Caaguazú, Repatriación (Paraguay)	25°32′16″ S	55°59′24″ W	[48]
Concepción, Azotey (Paraguay)	23°19′08″ S	56°29′16″ W	[48]
Junín, Chanchamayo (Peru)	11°03′00″ S	75°18′14″ W	[42]
Private field, Primero de Mayo, Colonia Benítez, Chaco (Argentina)	27°20′17″ S	58°58′01″ W	This publication
Los Chaguares, Primero de Mayo, Colonia Benítez, Chaco (Argentina)	27°19′59″ S	58°57′57″ W	This publication

**Table 2 insects-13-01069-t002:** Calibration and evaluation of ellipsoid models used to characterize the niche of *Lepturges limpidus*. Bold rows highlight the method selected to create the final model. Columns show evaluation metrics used for best-model selection: mean AUC, partial ROC, number of valid iterations, omission rate, mean values of prevalence in environmental (E-Space) and geographical space (G-space).

Method	Mean AUC Ratio at 5%	*p*-Value (Partial ROC)	Valid Iterations	Omission Rate	Prevalence in E-Space	Prevalence in G-Space
**covmat**	**1.984**	**<0.001**	**500**	**0**	**0.712**	**0.712**
mve1	1.68	<0.001	500	0	0.693	0.693

**Table 3 insects-13-01069-t003:** Analysis of ellipsoidal niche overlap based on environmental conditions relevant for *Lepturges limpidus* and *Catostemma fragrans*.

Bioclimatic Variables	Niche Volume: *C. fragans*	Niche Volume: *L. limpidus*	Overlap	Overlap (*p*-Value)	Size Ratio: Niche 1 vs. Niche 2	Size Ratio: Niche 2 vs. Niche 1
All	4.947	114.518	0.529	0.845	0.529	1.890
Precipitation	5.458	42.262	0.533	0.651	0.533	1.876
Temperature	0.664	32.309	0.488	0.936	0.488	2.050

## Data Availability

The data that supports the findings of this study are available in the Appendix A of this article.

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
