# Peer review of "New Distributional Records and Characterization of the Climatic Niche of Lepturges (Lepturges) limpidus Bates, 1872 (Coleoptera, Cerambycidae): Sink or Source Population?"

_insects, 2022, doi:10.3390/insects13111069_

Round 1

Reviewer 1 Report

The present manuscript is devoted to occurrence of a longhorn beetle Lepturges limpidus in Argentina, providing an assessment of its ecological niche. 

The manuscript is written in rather clear English, but some minor improvements are required as demonstrated (see document; not all but only some corrections were made!).

NE part of Argentina is within the potential distribution range of Lepturges limpidus and this species is not in the list of invasive insects, therefore the authors are encouraged to avoid unjustified accent on "invasiveness" or "potential invasiveness" of this species.

The authors are encouraged to supplement their datasets used for the assessments by the museums material and not to limit themselves to GBIF and other on-line data portals. Museums collections contain billions of specimens provided with more or less detailed data including date of collecting and (possible) foodplants, both relevant to the present research. Using more data could potentially change the results (especially would it be discovered L. limpidus is not monophagous).

Several, mainly minor, corrections and comments to be found in the manuscript file. 

The manuscript is recommended for the publication after a minor revision.

Author Response

Dear Dr.

Many thanks for the reviews and constructive comments on our manuscript (insects-1945354), entitled “New distributional records and characterization of the climatic niche of Lepturges (Lepturges) limpidus Bates, 1872 (Coleoptera, Cerambycidae): sink or source population?”. Based on the reviewers’ comments we have modified the original manuscript and proofread it carefully.

In the following, we provide our specific responses to each comment raised by the reviewers. We thank the reviewers once again, and believe that the revised manuscript has improved. We hope that the new version meets the standards of Insects.

Sincerely,

Néstor G. Valle (on behalf of all authors)

Reviewer 2 Report

Taking into account the facts:

1. L. limpidus is widely distributed from central Mexico to southern Paraguay and new records from Argentina are only about 200 km away form the nearest previous site (obtained from literature). For insects (that do not respect political boundaries), it is a very short distance to fly, therefore, the appearance of this species in new areas should not be surprising.

2. As Authors wrote “Vegetation and environmental profile on new localities are similar to those within the previously known distribution of L. limpidus”, This factor favors the natural distribution of the species and natural expansion of its range. So once more - the appearance of this species in new areas should not be surprising.

3. New records concern only 5 specimens, so it is hard to consider in aspect of “population”. More intensive field studies should be carried out (with the use of special traps) in order to find a larger number of individuals.

4. No studies were carried out on the possible ontogenetic development of L. limpidus on new locations, conducting observations on the development of the species in field conditions, identifying larval stages, would give a dependable answer about the “stability of the population in the new area”

It should be seriously considered whether it is advisable to conduct that type of research presented in the paper and testing “whether these new records could represent established populations”, at this general very poor stage of L. limpidus study. The Authors are asked to prove that the general purpose of the research is valid and that there is a need for such research.

Moreover:

-Page 2. No information is provided about the species habitat preferences. Instead of occurrence records from literature and from new observations from Argentina precise data on habitats were the records come from is needed.

-Page 2. “As a first step towards predicting and preventing invasion, we seek to understand whether the new records in the northeast of Argentina could represent an established population of L. limpidus”. Since the new records are based on only 5 specimens, can we name that group of insect “population”?

-Native and non-native (secondary) range of L. limpidus could be clearly described and presented on the map/figures.

-Page 6. The Map is confusing, this part of the left is too small. The Authors should present: - a high suitable area, - a moderate suitable area, - low suitable area, - unsuitable areas.

Author Response

(The authors gave the same response as above.)

Reviewer 3 Report

The manuscript is focused on the economically important group of insects - longhorn beetles. Specifically, the authors model the ecological niche of Lepturges limpidus which was recently found outside its native range, in Argentine. The authors use the model to assess the status of new records, and evaluate how the distribution of the known host plant of the species limit the distribution of the beetle.

The manuscript is appropriate for Insects. The question that authors address is original and well-defined. The results provide an advancement of the current knowledge of the distribution of a potentially invasive species. The material and methods used are appropriate and well-described. The illustrations are good and comprehensive. The list of references is very comprehensive but perhaps some references are redundant (i.e. 7-11 are about various groups longhorn beetles not discussed in the present manuscript). The English is clear and the text is in general well-written but scientific names should be italicized throughout the text.

Author Response

(The authors gave the same response as above.)
